# Exploring the theory, barriers and enablers for patient and public involvement across health, social care and patient safety: a protocol for a systematic review of reviews

Josephine Ocloo,[1] Sarah Garfield,[2,3] Shoba Dawson,[4] Bryony Dean Franklin[2,3]

¹Health Service and Population Research Department, CIS, CLAHRC, Institute of Psychiatry, Psychology and Neuroscience (IoPPN), King's College London, London, UK
²Centre for Medication Safety and Service Quality, Imperial College Healthcare NHS Trust, London, UK
³University College London, School of Pharmacy, London, UK
⁴University College London Ear Institute, evidENT, London, UK

**Correspondence to**
Dr Josephine Ocloo;
josephine.ocloo@kcl.ac.uk

## ABSTRACT

**Introduction** The emergence of patient and public involvement (PPI) in healthcare in the UK can be traced as far back as the 1970s. More recently, campaigns by harmed patients and their relatives have emerged as a result of clinical failings in the NHS, challenging paternalistic healthcare, which have led to a new focus on PPI in quality and safety, nationally and internationally. Evidence suggests that PPI within patient safety is often atheoretical and located within a biomedical discourse. This review will explore the literature on PPI across patient safety, healthcare and social care to identify theory, barriers and enablers that can be used to develop PPI in patient safety.

**Methods and analysis** Systematic searches of three electronic bibliographic databases will be conducted, using both MeSH and free-text terms to identify empirical literature published from database inception to May 2017. The screening process will involve input from at least two researchers and any disagreement will be resolved through discussion with a third reviewer. Initial inclusion and exclusion criteria have been developed and will be refined iteratively throughout the process. Data extraction from included articles will be conducted by at least two researchers using a data extraction form. Extracted information will be analysed using a narrative review approach, which synthesises data using a descriptive method.

**Ethics and Dissemination** No ethical approval is required for this review as no empirical data were collected. We believe that the findings and recommendations from this review will be particularly relevant for an audience of academics and policymakers. The findings will, therefore, be written up and disseminated in international peer-reviewed journals and academic conferences with a health focus. They will also be disseminated to leading health policy organisations in the NHS, such as NHS England and NHS Improvement and national policy bodies such as the Health Foundation.

## INTRODUCTION

The importance of involving patients, service users, carers and the public in the

---

**Strengths and limitations of this study**

► This review will address a perceived knowledge and theory gap with PPI in patient safety.
► It will critically evaluate and synthesise evidence from across health, social care and patient safety.
► The review will not include the grey literature.
► The review will focus solely on published peer-reviewed reviews.

---

UK in health and social care and research has grown significantly in recent decades.[1–4] These developments have been linked to the growth in policy initiatives around citizenship, democracy and rights.[5] More recently, the idea of partnership working with patients and the public has gained even more prominence as a result of serious clinical and service failings in the UK,[6 7] and internationally.[8–10] These initiatives have often been driven by the campaigns of patients who have experienced harm and their relatives, leading to a new focus on patient and public involvement (PPI) and its importance in improving quality and safety within healthcare.

In the UK, a commitment to PPI is now firmly enshrined in key legislation. This covers the Health and Social Care Act,[11] the NHS Constitution[12] and the duty by NHS England (s13Q of the National Health Service Act 2006 (as amended by the Health and Social Care Act 2012)) to properly involve patients and the public in its commissioning processes and decisions. In addition, key regulations set out essential standards of quality and safety that people who use health and adult social care services have a right to expect.[13] These rights are underpinned by wider but complimentary policy approaches such as public and patient experience and engagement (PPE). These approaches aim

to place people who use services at the centre of care, to understand their experience of services, to empower them to make decisions and to involve them in the design and delivery of care.[14] The NHS next stage review identified three strands of quality: patient experience, patient safety and clinical effectiveness. PPE approaches aim to ensure that patient experience sits as an equal partner in these three strands of quality.[15]

These types of PPI initiatives are by no means unique to the UK and similar reforms in health and social care can be seen across a range of international settings.[16–20]

Despite these policy developments, there have been an increasing number of criticisms about the nature of PPI in practice. There is uncertainty about how to do it well, in ways that constitute genuine partnerships and which involve a diversity of patients and the public, rather than a few selected individuals.[21–23] In particular, in the area of patient safety, the literature on PPI has been seen as dominated by a biomedical approach,[24 25] has been atheoretical[26] and has not addressed power inequities and discrimination.[27] This has exposed PPI to criticisms of exclusivity and tokenism.[28]

This is seen as leading to a narrow model of PPI that fails to empower patients and the public in the involvement process. This gap is seen to be out of keeping with the wider literature on PPI more broadly in healthcare, which highlights the contested and bottom-up nature and drivers for involvement and the way in which various global health social movements have provided collective challenges to poor care and discriminatory or paternalistic services and medical policy and belief systems.[29] These drivers have led to the development of theory, methods and approaches, particularly within mental health, that have been used to develop wider PPI models and methods based upon coproduction and partnership. In patient safety, despite the campaigns by patients who have experienced harm and their relatives acting as a catalyst for the patient safety movement, there is evidence that suggests that lay members are struggling to influence decisions and are largely expected to work within existing systems in improving quality and safety.[7 26] Involvement at this level has, therefore, been criticised as providing little opportunity to influence decision-making processes in any depth, maintaining power differentials and the status quo.

This review is being undertaken to address a perceived knowledge and theory gap with PPI in patient safety. To address this, the review will identify reviews of the published empirical literature, to look at: 'What are the theories, barriers and enablers in undertaking patient and public involvement in health social care and patient safety'?.

## Terminology
There is considerable confusion about the use of terminology in this area. A number of different terms are often used synonymously with involvement, such as engagement or participation, while the terms patients and the public are also used interchangeably with 'citizen', 'consumer', 'layperson' or 'service user'. These conceptual differences have emerged from disparate traditions social movements, policies and practices to describe the involvement process.[30] They have also been used to imply a greater or lesser level of involvement, power or influence in decision-making processes within an organisation. However, this language does not always reflect the underlying ethos of these involvement activities.[31] In the absence of a consensus on terminology, we define involvement as an activity that is done 'with' or 'by' patients or members of the public rather than 'to', 'about' or 'for' them.[32] This definition reflects the fact that the involvement process has increasingly come to be seen as a process of partnership: '…the active participation of patients, carers, community representatives, community groups and the public in how services are planned, delivered and evaluated. It is broader and deeper than traditional consultation. It involves the ongoing process of developing and sustaining constructive relationships, building strong, active partnerships and holding a meaningful dialogue with stakeholders'. (p9)[13]

## Methods and analysis
### Study design
This systematic review of reviews will be conducted and reported in accordance with the Preferred Reporting Items for Systematic Review and Meta-Analysis Protocols (PRISMA-P) (see online supplementary file S1)[33] and the results reported following the PRISMA guidelines.[34] The flow diagram (figure 1) demonstrates the different stages of the systematic review and how these interact and influence one another.

### Study registration
On the basis of the PRISMA-P guidelines, this review was registered with the International Prospective Register of Systematic Reviews (PROSPERO) on 28/4/2017 (registration number CRD42017067848: (http://www.crd.york.ac.uk/PROSPERO/).

### Eligibility criteria
Studies will be included in this review if they fulfil the following criteria:

### Inclusion criteria
1. Type of study: systematic reviews that focus on the concept of, or approaches to, PPI and/or PPE across patient safety, healthcare and social care.
2. Setting: any organisational setting (eg, primary care, mental health, hospital, tertiary care, voluntary, etc).
3. Type of involvement: this review will focus largely on the collective level or what has also been referred to as public involvement (this literature generally relates to public involvement in strategic decisions in health services, eg, in service improvement planning, and/or organisational design, and can cover various ar-

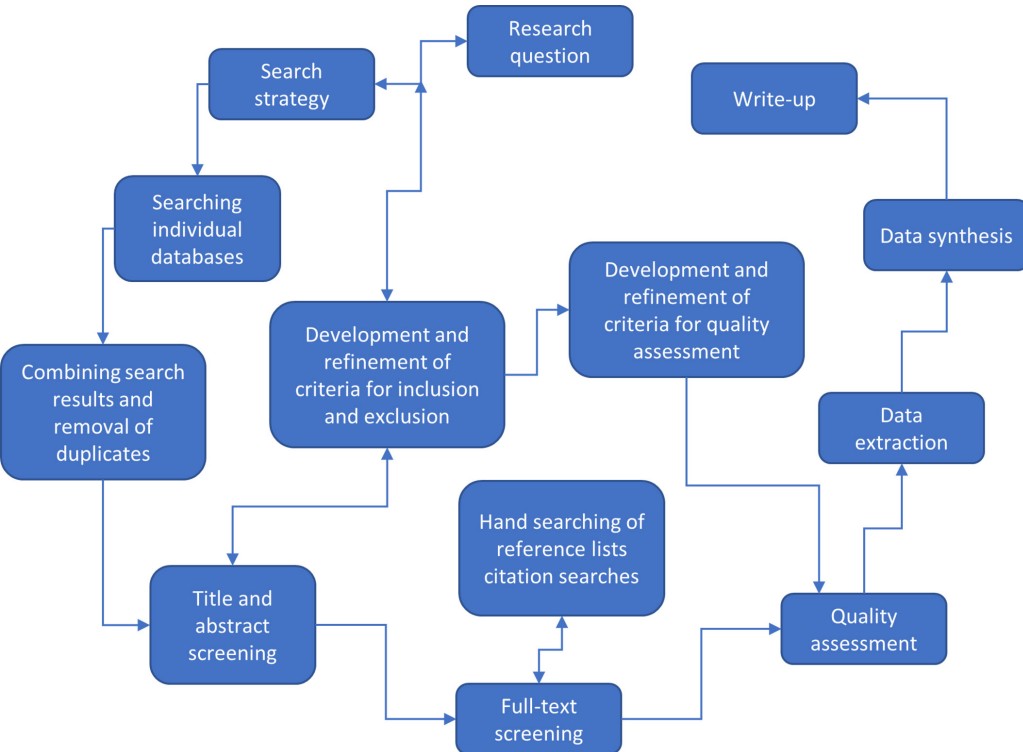

**Figure 1** Flow diagram—stages of the systematic review.

eas at a local or national level in governance, policy making, commissioning, monitoring, evaluation and research). The review will, however, look at some examples of involvement in direct care, but only where this relates to activities to improve health, social care or patient safety and quality more widely. The literature covering public involvement is distinct from literature focusing on patient involvement, which refers more specifically to 'the involvement of individual patients, together with health professionals, in making decisions (including shared decision making) about their own care.[35] The review will not include this much wider and more substantial body of literature on aspects or proxies for patient involvement, or engagement in their own clinical treatment such as shared decision making and patient centredness.

4. Study design: systematic reviews based on either published empirical studies (eg, using quantitative, qualitative or mixed methods) or theoretical papers. Where papers include both a systematic review and an empirical study, we will include data relating to the review if it is presented separately.

### Articles that meet any of the following criteria will be excluded

1. Systematic reviews that do not have a specific focus on PPI at the collective service improvement level.
2. Empirical studies.
3. Non-systematic reviews.
4. Reviews that focus on PPI at the individual level in treatment and decision making.
5. Reviews not written in English.

### Search strategy

Three main electronic bibliographic databases including MEDLINE, EMBASE and PsycINFO will be searched for potential studies from the inception of databases to May 2017 as described in table 1.

A comprehensive search strategy will include a combination of five main blocks of terms including and relating to public involvement (public, patient, carer, consumer, citizen, lay, service user, stakeholder, family, relative, survivor), type of involvement (involvement, collaboration, engagement, partnership, consultation, participation, user-led, consumer or patient panel, advisory board/group/panel), health and social care setting (health services, health care, social care, public health, mental health, etc), patient safety (safe*, adverse safety (safe*, adverse event$, error*, etc) and type of review (systematic, narrative, meta, bibliometric); using a combination of Medical Subject Headings (MeSH) terms and free text (see online supplementary file S2: eg, search strategy). Terms used to describe public involvement and type of involvement were similar to other studies.[36 37] Database searches will be supplemented by citation searches and reference lists of included studies. In addition, several scoping exercises in these electronic databases will be applied to maximise the sensitivity and specificity of the developed search strategy.

### Screening of studies

An Endnote library will be used to combine and export the results of the searches from different databases. Duplicates will be removed prior to the selection of studies. Study selection will be completed in two stages. First, titles

**Table 1** A structured search of key electronic bibliographic databases

| Search type | Database | Database history | Filters applied |
|---|---|---|---|
| Systematic review of the literature | MEDLINE (via OVID 1946 to May 2017) | Biomedical database produced by the National Library of Medicine, with over 3000 journals | Limited to systematic reviews, narrative reviews, bibliometric reviews and meta-analysis |
| | EMBASE (1974 to May 2017) | Biomedical and pharmacological database, with a more European focus | Limited to systematic reviews, narrative reviews, bibliometric reviews and meta-analysis |
| | PsycINFO (1967 to May 2017) | Produced by the American Psychological Association covering over 1300 journals from Psychology and related disciplines | Limited to systematic reviews, narrative reviews, bibliometric reviews and meta-analysis |

and abstracts will be screened in order to identify eligible and relevant studies, followed by full-text screening to identify reviews eligible for inclusion.

### Abstract and title screening

Abstracts and titles will be used to initially screen and identify eligible and relevant studies using the selection criteria above for the articles remaining in the literature review. A random sample of 10% titles and abstracts will be independently screened by both the lead researcher (JO) and a secondary reviewer (SG or SD). Any disagreements will be resolved through discussions until consensus is reached. However, if we are unable to reach consensus, disagreements will be resolved through discussions with an independent third reviewer (BDF). Once a consensus is reached for the 10% sample, the remaining titles and abstracts will be screened by the lead researcher alone.

### Full-text screening

At the outset of this process, a 10% sample of full-text papers will be screened and reviewed independently by both the lead researcher and a second reviewer (SG or SD). Any disagreements will be resolved through discussion or involving a third reviewer. If necessary, a further 10% of papers will be independently reviewed to test for agreement and then the remaining reviews will be screened by the lead researcher (JO) alone. Along with the lead researcher, SG and SD will screen all included papers to ensure that they meet the inclusion criteria.

### Data extraction and synthesis

A data extraction form will be used (see online supplementary file S2) to focus on the characteristics that are relevant to this study. The form will be piloted on 10% of randomly selected studies. Any necessary modifications will then be made to the data extraction tool. Data from the included studies will be extracted and categorised using the following headings below.

► Study characteristics: authors, year, study aims, type of study/study design, health topic focus and setting.
► Participant characteristics: types of people involved (including age, ethnicity, gender, etc).

► Evidence of equality and diversity (in accordance with the Equality and Human Rights Act and NHS Constitution).
► Definition(s) of PPI.
► Methods of PPI (involvement can be seen as a spectrum, with a range of service user involvement activities that can take place at multiple levels ie, consultation, collaboration or user control).[38 39]
► Use of theory (to understand, analyse, describe and/or facilitate PPI activities).
► Barriers and facilitators to involvement.
► Reported impacts of PPI.

The data extraction process will be carried out by both the lead researcher and a secondary reviewer (SG or SD) as per the earlier processes. Any disagreements will be resolved through discussion or involving a third reviewer.

A systematic narrative approach will be used for synthesising the results of this review due to the heterogeneous nature of the included published empirical literature.[40] This will include familiarisation with the papers and identifying data relating to the themes linked to study aims and outcomes. Data will be summarised in a descriptive form in order to allow us to draw conclusions about the available evidence.

### Quality assessment

A quality assessment is normally conducted in systematic reviews using quality assessment checklists such as Critical Appraisal Skills Programme[i] to assess the quality of individual studies. As this is a systematic review of reviews, an empirically developed instrument for documenting the methodological quality of systematic reviews, Assessment of Methodological Quality of Systematic Reviews)[ii], will be used in this study to determine whether eligible reviews meet the minimum criteria based on quality.

---

[i]http://www.casp-uk.net/casp-tools-checklists
[ii]https://amstar.ca/Amstar_Checklist.php

## ETHICS AND DISSEMINATION

No ethical approval is required for this review as it is a systematic review and does not involve the collection of primary data.

We believe the findings and recommendations from this review will be particularly relevant for an audience of academics and policymakers given its focus across different sectors of healthcare. The findings will therefore be written up and published in international peer reviewed journals as well as disseminated through academic conferences with a health focus inline with suggestions made by the Guidance for Reporting Involvement of Patients and the Public (GRIPP2).[41]

The review, however, will also be used to provide opportunities for testing methods of PPI in practice in patient safety and evaluating its impact in improving services. The findings will, therefore, be disseminated via a range of policy organisations such as NHS England and NHS Improvement as well as national policy bodies such as the Health Foundation and its Q Network. Q is an initiative connecting people, who have improvement expertise, across the UK. It is led by the Health Foundation and supported and cofunded by NHS Improvement.

**Contributors** JO developed the intellectual idea for the review and led on all of the work in drafting the protocol and its various components. SG and SD (second and third authors) provided detailed input concerning the study design and methods and analysis and SD in addition to this and alongside JO developed the search strategy. BDF contributed to the intellectual development of the protocol, commenting on ideas, drafts and in making suggestions and acted as an independent person in helping to resolve disagreement and reach consensus. JO, SG, SD and BDF were all involved in the drafting of the protocol and approval for publication.

**Funding** The Health Foundation, National Institute for Health Research (NIHR).

**Disclaimer** The research is supported by the Centre for Implementation Science (CIS) which is part of the National Institute for Health Research (NIHR) Collaboration for Leadership in Applied Health Research and Care South London at King's College Hospital NHS Foundation Trust and the National Institute for Health Research (NIHR) Imperial Patient Safety Translational Research Centre. The views expressed are those of the author(s) and not necessarily those of the NHS, the NIHR or the Department of Health.

**Competing interests** None declared.

**Provenance and peer review** Not commissioned; externally peer reviewed.

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
