## [Reviewer comments · BMJ Open]

ARTICLE DETAILS

TITLE (PROVISIONAL)	Exploring the theory, barriers and enablers for patient and public involvement across health, social care and patient safety: A protocol for a systematic review of reviews.
AUTHORS	Ocloo, J; Garfield, Sarah; Dawson, Shoba; Dean Franklin, Bryony

VERSION 1 – REVIEW

REVIEWER	Dr Jonathan Boote University of Sheffield, UK
REVIEW RETURNED	12-Jul-2017

GENERAL COMMENTS	Well written protocol for a review of reviews on the topic of ppi in health, social care and patient safety. I have no suggestions for improving the paper.
---

REVIEWER	Sebastian S Fuller St George's University of London
REVIEW RETURNED	28-Jul-2017

GENERAL COMMENTS	This is a very interesting article describing the protocol for a systematic review of reviews on the subject of PPI and patient safety. I am not an expert on systematic reviews, but to my knowledge this protocol highlights good methodology for the planned review. When undertaken, this review may highlight best practices for future PPI work in this field, which is necessary, as the authors mention the dearth of systematic and robust approaches to PPI in patient safety. While this article is generally well written, there are some syntax issues that should be corrected prior to publication: Page 3, lines 15-16: this sentence is in the past tense but should be in the future tense. Page 6: the lead researcher is first identified on line 36; if identifying initials for this person are used it should be here (at the first mention), rather than on line 49 (where it is currently). Page 7: the last sentence could be edited for clarity and flow. Additionally, on page 5, from line 13: Please define "collective involvement" more clearly - I am not sure I fully grasp the differentiation made between collective, direct care, and shared decision making. I am not sure what precisely is being defined here, please clarify this objective.
---

REVIEWER	Professor David Evans Department of Health and Social Sciences University of the West of England UK
REVIEW RETURNED	17-Aug-2017

GENERAL COMMENTS	This is a protocol for a review of review on PPI which I think could make a useful contribution, and so I would be pleased to see the protocol published if the various issues below can be addressed.  1. Introduction: the discussion of the relationship of the term 'patient and public involvement' to other terms like 'patient experience and engagement' needs to be developed. At the moment there is a rather simplistic statement that the latter term is a 'wider but complimentary [sic] policy' approach. Rather the reality is that there is a terminological and conceptual muddle over these terms with different institutions and authors using terms like involvement, engagement and participation sometimes in synonymous and sometimes in contradictory ways. The quote at the bottom of page 3 actually exemplifies this. The Introduction needs to reflect this complexity in a more informed way. 2. The term 'patient representative' (p4) is problematic and I suggest should be avoided as it is widely acknowledged that being representative of other patients is not realistic or meaningful. Terms like 'public contributor' or 'lay member' are advised. 3. Critically, the protocol states an aim but not a research question, which should also be included. The PRISMA-P checklist recommends that this should follow the PICO framework, which is not done here, despite the appended checklist indicating that this item in the checklist has been met. 4. Search strategy: reference could usefully be made to the recent publication: Rogers M, Bethel A, Boddy K. Developing and testing a MEDLINE search filter for identifying patient and public involvement in health research. Health Info Libr J. 2016;34:125–133. https://doi.org/10.1111/hir.12157. 5. Dissemination is not discussed, and it would be good to plan to adhere to the recently published GRIPP 2 guidelines: Staniszewska et al. GRIPP2 reporting checklists: tools to improve reporting of patient and public involvement in research Research Involvement and Engagement (2017) 3:13DOI 10.1186/s40900-017-0062-2 6. Appendix 2: Data extraction form - Definition(s) of PPI - ground breaking though Arnstein was, this is a very old reference and I would say a superseded basis for defining PPI. PPI is more accurately seen as multidimensional, see for example: Gibson et al. 2017 Evaluating patient and public involvement in health research: from theoretical model to practical workshop. Health Expectations DOI: 10.1111/hex.12486 Finally two very minor points:  7. In a couple of places - abstract and main text - it talks about learning from the wider public sector, but in fact this is narrowly focused on health and social care. 8. Strengths and limitations box - first sentence makes a claim for the contribution of the protocol which would actually be the contribution of the substantive review. Although I think all these points need to be addressed, none of them should be too difficult, so I think with minor amendments this could usefully be published.
--

VERSION 1 – AUTHOR RESPONSE

Reviewer 1

Comment: Well written protocol for a review of reviews on the topic of ppi in health, social care and patient safety. I have no suggestions for improving the paper.

Response: We are pleased the reviewer found the protocol well written.

Reviewer 2

Comment: Page 3, lines 15-16: this sentence is in the past tense but should be in the future tense.

Response: This point has now been superseded through editing of this section

Comment: Page 6: the lead researcher is first identified on line 36; if identifying initials for this person are used it should be here (at the first mention), rather than on line 49 (where it is currently).

Response: This has now been superseded via the removal of the conclusion to conform to journal requirements.

Comment: Page 7: the last sentence could be edited for clarity and flow.

Response: This has now been addressed with the initials of the first author included (pg. 5, title and abstract screening).

Comment: Additionally, on page 5, from line 13: Please define "collective involvement" more clearly - I am not sure I fully grasp the differentiation made between collective, direct care, and shared decision making. I am not sure what precisely is being defined here, please clarify this objective.

Response: The inclusion criteria has been rewritten to clarify what collective involvement is and how it is different from direct care and shared decision-making (pg. 4, type of involvement).

Reviewer 3

Introduction: the discussion of the relationship of the term 'patient and public involvement' to other terms like 'patient experience and engagement' needs to be developed. At the moment there is a rather simplistic statement that the latter term is a 'wider but complimentary [sic] policy' approach. Rather the reality is that there is a terminological and conceptual muddle over these terms with different institutions and authors using terms like involvement, engagement and participation sometimes in synonymous and sometimes in contradictory ways. The quote at the bottom of page 3 actually exemplifies this. The Introduction needs to reflect this complexity in a more informed way.

Response: We have addressed these points by recognising the terminological confusion, clarifying that we have decided to use the term 'involvement' and have defined what we mean by this (pg. 3, Terminology)

Comment: The term 'patient representative' (p4) is problematic and I suggest should be avoided as it is widely acknowledged that being representative of other patients is not realistic or meaningful. Terms like 'public contributor' or 'lay member' are advised.

Response: We have replaced the term patient representative to use the term lay members (pg. 3, para 1).

Comment: Critically, the protocol states an aim but not a research question, which should also be included.

Response: We have now amended the protocol to include a research question as suggested by the reviewer (Aim of the review and research question, pg. 3, para 2)

Comment: The PRISMA-P checklist recommends that this should follow the PICO framework, which is not done here, despite the appended checklist indicating that this item in the checklist has been met.

Response: We agree with the reviewer that this is potentially misleading.

As this paper does not focus on any particular intervention, there is no comparator. Therefore, PICO framework was not relevant. This has now been clarified in the PRISMA checklist (Appendix page 2, objectives)

Comment: Search strategy: reference could usefully be made to the recent publication: Rogers M, Bethel A, Boddy K. Developing and testing a MEDLINE search filter for identifying patient and public involvement in health research. Health Info Libr J. 2016;34:125–
<https://emea01.safelinks.protection.outlook.com/?url=https%3A%2F%2Fdoi.org%2F10.1111%2Fhir.12157&data=01%7C01%7Cjosephine.ocloo%40kcl.ac.uk%7Cf783a880ffc4d4fa33608d4e5853e92%7C8370cf1416f34c16b83c724071654356%7C0&sdata=QcKGdVHxZI0X5CsxQR2rCzHWchejMR%2B%2FU%2BWx1BMBLc0%3D&reserved=0>

Response: Thank you for pointing this out. This reference has now been added to reflect the similarities between the search strategies (p.5, para 1).

Comment: Dissemination is not discussed, and it would be good to plan to adhere to the recently published GRIPP 2 guidelines: Staniszewska et al. GRIPP2 reporting checklists: tools to improve reporting of patient and public involvement in research Research Involvement and Engagement (2017) 3:13DOI 10.1186/s40900-017-0062-2

Response: Thank you for pointing this out. This reference has now been added to reflect the similarities between the search strategies (p.5, para 1).

Comment: Dissemination is not discussed, and it would be good to plan to adhere to the recently published GRIPP 2 guidelines: Staniszewska et al. GRIPP2 reporting checklists: tools to improve reporting of patient and public involvement in research Research Involvement and Engagement (2017) 3:13DOI 10.1186/s40900-017-0062-2

Response: We have included the section on dissemination and this reference to show how the findings will be disseminated in line with GRIPP2 (pg. 7, para 2, Ethics and dissemination section)

Comment: Appendix 2: Data extraction form - Definition(s) of PPI - ground breaking though Arnstein was, this is a very old reference and I would say a superseded basis for defining PPI. PPI is more accurately seen as multidimensional, see for example: Gibson et al. 2017 Evaluating patient and public involvement in health research: from theoretical model to practical workshop. Health Expectations DOI: 10.1111/hex.12486

Response: We agree that subsequent critiques of Arnstein reflect the multi-dimensional nature of PPI and we have therefore amended the wording in the data extraction form to reflect this and added more up to date references including the Gibson reference suggested by the reviewer (pg. 12, Appendix 2)

Comment: In a couple of places - abstract and main text - it talks about learning from the wider public sector, but in fact this is narrowly focused on health and social care.

Response: We have now removed the reference to the wider public sector

Comment: Strengths and limitations box - first sentence makes a claim for the contribution of the protocol which would actually be the contribution of the substantive review.

Response: This point has now been superseded by editing of this section and the removal of the sentence (pg. 1).

VERSION 2 – REVIEW

REVIEWER	Sebastian S Fuller St George's University of London
REVIEW RETURNED	15-Sep-2017

GENERAL COMMENTS	Issues identified in the first review have been dealt with by the authors adequately. The article is much improved and I am happy to recommend that it be published.
--

REVIEWER	David Evans University of the West of England, UK
REVIEW RETURNED	26-Sep-2017

GENERAL COMMENTS	You have satisfactorily addressed all of my issues from the previous iteration and I am now happy to recommend publication.
---